# Drought Resistance and Ginsenosides Biosynthesis in Response to Abscisic Acid in *Panax ginseng* C. A. Meyer

**DOI:** 10.3390/ijms24119194

**Published:** 2023-05-24

**Authors:** Lingyao Kong, Peng Chen, Cheng Chang

**Affiliations:** College of Life Sciences, Qingdao University, Qingdao 266071, China

**Keywords:** ginseng, abscisic acid, drought, ginsenoside, leaves, roots

## Abstract

Drought stress adversely affects the production of the perennial medicinal herb *Panax ginseng* C.A. Meyer. Phytohormone abscisic acid (ABA) regulates many processes in plant growth, development, and response to environments. However, whether drought resistance is regulated by ABA in *Panax ginseng* remains unknown. In this study, we characterized the response of drought resistance to ABA in *Panax ginseng*. The results showed that the growth retardation and root shrinking under drought conditions in *Panax ginseng* were attenuated by exogenous ABA application. Spraying ABA was shown to protect the photosynthesis system, enhance the root activity, improve the performance of the antioxidant protection system, and alleviate the excessive accumulation of soluble sugar in *Panax ginseng* under drought stress. In addition, ABA treatment leads to the enhanced accumulation of ginsenosides, the pharmaceutically active components, and causes the up-regulation of *3-hydroxy-3-methylglutaryl CoA reductase* (*PgHMGR*) in *Panax ginseng*. Therefore, this study supports that drought resistance and ginsenosides biosynthesis in *Panax ginseng* were positively regulated by ABA, providing a new direction for mitigating drought stress and improving ginsenosides production in the precious medicinal herb.

## 1. Introduction

In the natural environment, plants are subject to various stresses such as water deficit (drought), salinity, extreme temperatures, and pathogen infections [1,2]. As one of the most prevalent abiotic stresses, drought usually causes osmotic stress and leads to water loss from plant tissues, adversely affecting plant growth and development [3,4]. During adaptation to stressful environments, plants have evolved multifaceted mechanisms, including closing stomatal, reducing photosynthesis, enhancing root activity, scavenging reactive oxygen species (ROS), and accumulating osmolytes to cope with drought conditions [5,6,7,8]. The phytohormone abscisic acid (ABA) plays an important role in the plant response to drought stress [9,10,11,12]. ABA could promote seed dormancy to surpass the drought conditions, and the seed will germinate when the soil water conditions are suitable for growth [9]. ABA also promotes stomatal closure under drought conditions to prevent water loss [12]. In addition, the biosynthesis of ABA was enhanced in response to the drought stress in various plant species [2].

*Panax ginseng* C.A. Meyer is a perennial herbaceous plant from the family *Araliaceae*. As a medicinal herb, *Panax ginseng* has been cultivated for over 2000 years in Northeast China for its highly valued roots and rhizomes [13,14]. Ginsenosides are isolated from *Panax ginseng* as the most important pharmaceutically active components [15,16,17,18]. Up to now, more than 150 kinds of natural ginsenosides have been identified from *Panax ginseng* [19,20]. Most of these ginsenosides belong to the triterpene saponins, which are composed of aglycone and glycosyl. Based on the difference among aglycones, ginsenosides could be divided into three groups [19,20]. The first group was protopanaxadiol-type saponins, in which glycosyl was attached to the β-OH at C-3 and/or C-20, such as ginsenosides Rb1, Rb2, Rc, Rd, Rg3, and Rh2 [19,20]. The second group also was protopanaxadiol-type saponins, in which glycosyl was attached to the α-OH at C-6 and/or β-OH at C-20, such as ginsenosides Re, Rf, Rg1, Rg2, and Rh1 [19,20]. These two types of saponin contain a dammarane skeleton, which is the main pharmacological active ingredient [19,20]. The third group was oleanane-type pentacyclic triterpenoid saponins, and only ginsenoside Ro was isolated currently, which was the minimum content in the ginsenoside [21].

During its long (4 to 6 years) growth period, *Panax ginseng* is subject to various environmental challenges [22,23]. It was reported that the relative soil water content optimal for the growth of *Panax ginseng* is 60–80%. Drought stress could damage the growth of *Panax ginseng* [22,23]. Developing new strategies to mitigate drought stress is essential for securing *Panax ginseng* production. Although ABA has been reported to regulate drought resistance in model and crop plants, whether drought resistance and ginsenosides biosynthesis are regulated by ABA in *Panax ginseng* remains unknown. In this study, we examined the response of drought resistance and ginsenosides biosynthesis to ABA in *Panax ginseng* and found that drought resistance and ginsenosides biosynthesis were significantly enhanced by ABA treatment. These results paved a new path for mitigating drought stress in the precious medicinal herb *Panax ginseng*.

## 2. Results

### 2.1. Drought Resistance Is Enhanced by ABA in Panax ginseng

To characterize the potential regulation of drought resistance by ABA in *Panax ginseng* C.A. Meyer, 3-year-old *Panax ginseng* seedlings were treated by stopping plant watering either without ABA (drought) treatment or with 15µM ABA spraying (i.e., as drought plus ABA (drought + ABA) treatment). After 35 days, the *Panax ginseng* seedling leaves from the drought treatment group showed significant wilting, whereas *Panax ginseng* leaves from the drought + ABA group remained turgid (Figure 1A). Then, we analyzed the phenotype of *Panax ginseng* roots from drought treatment and drought + ABA treatment groups. The epidermis of *Panax ginseng* roots from the drought treatment group showed serious shrinkage, and the treatment with drought + ABA did not cause a significant change in the epidermis phenotype of *Panax ginseng* roots (Figure 1B). To eliminate the potential effect of ABA on different soil moisture, we measured the soil moisture. As shown in Figure 1C, groups of drought treatment and drought + ABA treatment showed no obvious difference in soil water content (Figure 1C). As a result, this evidence suggested that exogenous ABA application significantly enhanced drought resistance in *Panax ginseng*.

### 2.2. Stomatal Closure under Drought Stress Is Promoted by ABA in Panax ginseng

It is widely accepted that ABA could promote the closure of the opening stomatal and inhibit the opening of the closure stomatal, thereby controlling moisture and gas exchange of plant tissues with the surrounding environments [24,25,26,27,28,29,30,31]. After treatment of *Panax ginseng* with drought or drought + ABA for 28 days, the stomatal movement was examined, and the apertures were measured under a microscope. The results showed that drought stress could significantly induce the stomatal closure in *Panax ginseng* leaves (Figure 2A). The pore size of *Panax ginseng* stomata from the drought + ABA treatment group was obviously smaller than those from the drought treatment group (Figure 2B). The results showed that stomatal closure under drought stress is improved by ABA treatment in *Panax ginseng*.

### 2.3. Physiological Characteristics of Panax ginseng under Drought Stress Is Affected by ABA Treatment

We then examined the physiological changes of *Panax ginseng* under drought conditions. First, we checked the chlorophyll content to analyze the effect of drought stress on photosynthesis in *Panax ginseng*. As shown in Figure 3A, drought treatment significantly reduced the chlorophyll content in the *Panax ginseng* leaves but has no effect on the chlorophyll content in *Panax ginseng* leaves sprayed with ABA, suggesting that ABA application could protect the photosynthesis system of *Panax ginseng* under drought conditions. Root activity is another indicator of plant growth status under stress conditions. As shown in Figure 3B, drought stress could enhance the root activity in *Panax ginseng*, and the *Panax ginseng* root activity from the drought + ABA treatment group was stronger than those under the drought treatment, suggesting that root activity of *Panax ginseng* under drought conditions is enhanced by ABA treatment.

During the evolution to adapt to the changing environment, plants acquired an effective system to scavenge free radicals and avoid the ROS damage to cells. Free radical scavenging enzymes include superoxide dismutase (SOD), catalase (CAT), and peroxidase (POD). We analyzed the activity of these enzymes in *Panax ginseng* under drought conditions and found that SOD and CAT activities were increased under drought condition, but the activity of *Panax ginseng* enzymes from the drought + ABA treatment group was higher than those from the drought treatment group (Figure 3C,D). These results indicated that exogenous ABA could improve the performance of the antioxidant protection system in *Panax ginseng*.

Malondialdehyde (MDA) level represents the degree of cytoplasmic membrane peroxidation. The higher MDA content indicates a higher degree of cell peroxidation and more damage to cells [32]. Therefore, we examined the MDA content in the leaf cells of *Panax ginseng*. As shown in Figure 3E, MDA content in *Panax ginseng* leaf cells from the drought + ABA treatment group was lower than those under drought treatment. These results indicated that exogenous ABA application could reduce the level of malondialdehyde in lipid membranes and avoid damage to cells by peroxidation in *Panax ginseng*.

Soluble sugar is the main osmotic adjustment substance in cells, and its content was increased under stress conditions. As shown in Figure 3F, drought treatment led to the increased content of soluble sugar in *Panax ginseng*, but the content of soluble sugar descended under drought + ABA treatment (Figure 3F). These results suggested that exogenous ABA treatment alleviated the excessive accumulation of soluble sugar in *Panax ginseng* under drought conditions.

### 2.4. The Accumulation of Ginsenoside under Drought Stress Was Enhanced by ABA Treatment in Panax ginseng Roots

Ginsenoside is the most valuable active component in *Panax ginseng*. We analyzed the content of ginsenoside in *Panax ginseng* roots under drought or drought + ABA treatment by employing high-performance liquid chromatography (HPLC). The results showed that both monomer and total ginsenoside were greatly increased in *Panax ginseng* roots under drought treatment and drought + ABA treatment (Figure 4). Interestingly, the content of ginsenoside in *Panax ginseng* roots under drought + ABA treatment is higher than those under drought without ABA treatment (Figure 4). These results indicated that the ginsenoside accumulation in *Panax ginseng* roots is enhanced by ABA treatment under drought conditions.

### 2.5. ABA Treatment Induced PgHMGR1/2 Expression in Panax ginseng

3-hydroxy-3-methylglutaryl CoA reductase (HMGR) is known as a rate-controlling enzyme of ginsenoside biosynthesis in *Panax ginseng* [21]. To examine the potential regulation of *PgHMGR* expression by ABA treatment in *Panax ginseng*, we treated the *Panax ginseng* with ABA and examined the *PgHMGR* expression using qRT-PCR. As shown in Figure 5, the expression of *PgHMGR1* and *PgHMGR2* was induced by ABA treatment, suggesting that upregulation of *PgHMGR1* and *PgHMGR2* might contribute to the ginsenoside accumulation enhanced by ABA treatment in *Panax ginseng* under drought conditions.

## 3. Discussion

Multifaceted roles of phytohormone abscisic acid in plant responses to drought stress have been well studied in model and crop plants. Under drought stress, ABA biosynthesis is potentiated, and accumulated ABA could promote stomatal closure, reduce water evaporation, and enhance plant tolerance to drought [33]. Our results demonstrated that the growth retardation and root shrinking caused by drought stress in *Panax ginseng* were compromised by ABA treatment. In addition, exogenous ABA application was shown to protect the photosynthesis system, enhance root activity, improve the performance of the antioxidant protection system, and alleviate the excessive accumulation of soluble sugar in *Panax ginseng* under drought conditions. Ginsenosides biosynthesis and *PgHMGR* expression are also enhanced by ABA treatment. These results clearly support that drought resistance and ginsenosides biosynthesis in *Panax ginseng* were promoted by ABA.

Chlorophyll is the main pigment essential for plant photosynthesis, and the content of chlorophyll is closely related to activity of plant photosynthesis [34]. Under drought conditions, the plant chloroplasts undergo a series of morphological and structural changes such as swelling and irregular arrangement, leading to a reduction in chlorophyll content. In addition, the ROS burst induced during drought stress could attack photosynthetic organs, and adversely affected the activity of photosynthetic membranes [24]. The results from this study proved that drought stress reduced chlorophyll content in leaves of *Panax ginseng*, while exogenous spraying of ABA inhibited the degradation of chlorophyll under drought and protected *Panax ginseng* leaves from further injury under drought stress.

ROS are produced when plants are undergoing aerobic metabolism. Abiotic stresses such as drought conditions could alter the metabolic balance and lead to the excessive accumulation of ROS [35]. Generally, excessive ROS caused damage to cell membrane system and destruction of photosynthesis [36]. SOD and catalase CAT are the main protective enzymes in plants against reactive oxygen species and play an important role in resisting a variety of environmental stresses, [37,38,39,40]. Therefore, the activity of SOD and CAT could be employed as a physiological index of plant stress physiology and senescence. The results of this study showed that the activities of SOD and CAT in the ABA-sprayed *Panax ginseng* leaves were significantly higher than those under drought treatment. The enhanced antioxidant enzyme activity contributes to the improved plant capacity to scavenge reactive oxygen species, thereby reducing the lipid peroxidation of membranes and protecting the integrity of membrane structures. These results were consistent with the fact that ABA induced the expression of plant antioxidant enzyme genes under stresses [41,42]. In addition, ROS accumulated under stress induces peroxidation of unsaturated fatty acids in membrane lipids to produce MDA, which could react with enzyme protein to deform the membrane structures. Our results showed that under drought, the MDA content in *Panax ginseng* was increased significantly, but the content of MDA remains unchanged under drought conditions after ABA was supplemented. These results suggested that ABA could relieve the injury caused by membrane lipid peroxidation under drought stress in *Panax ginseng*. Soluble sugar, an important osmotic regulator in plants, plays a vital role in plant adaptation to environmental stresses. Previous research has shown that soluble sugar plays a significant role in maintaining plant protein stability under drought stress. Our results displayed that drought stress induced the synthesis of soluble sugar in ginseng leaves, which was reduced by exogenous ABA. Therefore, ABA may be involved in the regulation of soluble sugar metabolism during drought stress.

In addition, this study showed that the accumulation of ginsenosides was significantly enhanced by ABA treatment in *Panax ginseng*. Another recent study demonstrated that drought stresses promoted ginsenosides accumulation and expression of ginsenoside biosynthesis genes in *Panax ginseng*. At the same time, the ABA accumulation and signaling in *Panax ginseng* were also potentiated by drought stress. These findings, together with results from this study, strongly support the positive regulation of ginsenosides biosynthesis by drought stress and ABA in *Panax ginseng*. *PgHMGR1* and *PgHMGR2* are the key regulatory enzymes of ginsenoside biosynthesis, and they are predominantly expressed in roots [13,21]. This study indicated that expression of *PgHMGR1* and *PgHMGR2* were significantly enhanced by ABA treatment. Previous studies have found that overexpressing *PgHMGR1* in ginseng adventitious roots increased the ginsenoside content [21,43]. Therefore, ABA might govern the ginsenoside biosynthesis by directly regulating the expression of *PgHMGR1* and *PgHMGR2*. These results provide a new avenue for enhancing drought resistance and ginsenosides production in the precious medicinal herb *Panax ginseng*.

## 4. Materials and Methods

### 4.1. Ginseng Materials and Growth Conditions

Three-year-old ginseng (*Panax ginseng* C.A. Meyer) seedlings used in this study were obtained from Tonghua, Jilin province, China. The dormant ginsengs were purchased and soaked in gibberellin acid (Sigma-Aldrich, St. Louis, MO, USA) for 4 h. Then, the ginsengs were grown in forest soil and vermiculite (1:1) and were kept under 50 μM m^−2^ s^−1^ light at 22 °C and 16 h light/8 h dark for 4 weeks in the greenhouse.

### 4.2. Drought Treatment, Drought + ABA Treatment, and Stomatal Aperture Measurement

For drought treatment, 3-year-old ginseng seedlings were grown under normal conditions for 4 weeks, and then water was withheld for 35 days. At the same time, after the water supply was stopped, the ginseng leaves were sprayed with 15 μM ABA (Sigma-Aldrich, St. Louis, MO, USA) every two days as drought + ABA treatment. The leaves and roots were collected for the subsequent experiments, and the phenotypic responses to drought or drought + ABA treatment were recorded. Three independent experiments were performed.

For stomatal aperture assay, epidermal strips were peeled from the ginseng leaves, which had grown for 28 days in drought and drought + ABA conditions. The chlorophyll on the epidermal strips was removed with a writing brush. The epidermal strips were immersed in MES buffer (10 mM MES-KOH (pH 6.15), 10 mM KCl, and 50 μM CaCl_2_) under light (90 μM/m^2^/s) for 2 h at 22 °C. Stomatal closure was photographed with an OLYMPUS BX53 microscope (Japan) and was measured with Image J 1.47 V software, with three epidermal strips per replicate and 30 stomata per strip.

### 4.3. Soil Moisture Content Analysis

The soil for cultivating ginseng was collected and weighed as m1. Then, soil samples were moved into the drying oven and dried at 105 °C for 6 h until the weight of the samples was no longer changed. The dried sample is weighed as m2. The soil moisture was calculated as follows: soil moisture content (%) = (m2 − m1)/m1 × 100%

### 4.4. Determination of Chlorophyll Content, Root Activity, MDA Content and Soluble Sugar Content

For the chlorophyll content measurement, chlorophyll was extracted by 80% acetone, and chlorophyll a and b content stand for total chlorophyll content. A spectrophotometer (SHIMADZU, Japan) was used to measure the sample’s absorbance at 665 nm and 649 nm. The content of chlorophyll a and b were calculated as follows: Ca (mg/L) = 13.95 × A665 − 6.88 × A649; Cb(mg/L) = 24.96 × A649 − 7.32 × A665 (Ca: the content of chlorophyll a; Cb: the content of chlorophyll b). The total chlorophyll content was calculated as follows: Chlorophyll content (mg/g) = ((Ca + Cb) × V × n)/m V: extraction volume (L); n: dilution-multiple; m: sample weight (g)

For root activity, 0.3 g ginseng root tip samples were soaked in 0.2%TTC (triphenyl tetrazlium chlorideat) at 37 °C for 2 h, then added to 1 M sulfuric acid, stopping the reaction. After the solution was removed, root tips were ground with ethyl acetate to obtain TTF (formazan). The extract demonstrated absorbance at 665 nm via spectrophotometer (SHIMADZU, Japan). The reduced amount of TTC is found from the standard curve. The reduction intensity of TTC (root activity) was calculated as follows: root activity (mg/(g.h)) = C/(m × t); C: reduction intensity of TTC (mg); m: root tip weight (g); t: hour (h).

The MDA content was determined by a reaction of MDA with 0.25% TBA (2-thiobarbituric acid) as described previously [32].

For soluble sugar content analysis, the ginseng leaves were ground in liquid nitrogen,; then, the content of sugar was determined as by Chen et al. [44].

### 4.5. Antioxidant Enzymatic Activity Analysis of SOD and CAT

To analyze the activity of antioxidant enzymes, fresh leaves (1 g) are ground in liquid nitrogen, and 5 mL 50 mM PBS (pH 7.8, containing 1 mM EDTA, 1 mM DTT, 2% *w*/*v* polyvinylpyrrolidone) was added to the powder. The homogenate was centrifuged at 15,000× *g* for 2 min at 4 °C. The supernatant was the crude extract of the antioxidant enzyme. The activity of superoxide dismutase (SOD, EC 1.15.1.1) was determined by the photochemical reduction of nitro blue tetrazolium (NBT) as described previously [37], with some modifications. In brief, the reaction mixture, containing 2 mL 50 mM PBS (pH 7.8), 1 mL 39 Mm methionine (Met), 1 mL 33 μM riboflavin, 1 mL 1.25 mM NBT, and 50 μL enzyme crude extract in a 10 mL transparent glass tube, was incubated for 20 min under 4000 Lux fluorescent. At the end of the reaction, samples demonstrates absorbance at 560 nm via UV–visible spectrophotometer (SHIMADZU, UV-2700). One enzyme unit is defined as the amount of enzyme required to inhibit NBT photochemical reduction by 50%, expressed as unit mg^−1^ protein.

The activity of catalase (CAT, EC 1.11.1.6) was determined by the ultraviolet absorption method as described previously [37]. H_2_O_2_ has a strong absorption at 240 nm wavelength. Catalase could decompose H_2_O_2_, and the absorption (A240) decreases with the reaction. The activity of catalase can be measured according to the change in absorption. A total of 3 mL reaction mixture, containing 2.65 mL PBS (pH 7.8), 300 μL 0.1 M H_2_O_2_, and 50 μL enzyme crude extract in a 5 mL transparent glass tube, demonstrates absorbance at 240 nm wavelength. The absorbance of H_2_O_2_ was recorded at 240 nm for 1 min. One enzyme unit is defined as one per min OD value decreased by 0.01. The enzyme activity was calculated by CAT activity (U/g min) = (ΔA240 × Vt)/(m × Vs × 0.01 × t); ΔA240: change of absorbance in reaction time; m: fresh weight of the sample (g); Vt: total volume of enzyme extraction solution (mL); Vs: the volume of enzyme solution used in the determination (mL); t: reaction time (min)

### 4.6. Determination of Ginsenosides Content by HPLC

The ginsenosides content was determined by HPLC, as described previously [13]. Ginseng root samples are dried at 40 °C and ground into powders. Samples of 0.5 g were soaked in 80% (V/V) methanol, ultrasonicated for 30 min, then stood overnight. Samples were re-ultrasonic extracted twice—30 each time with an interval of 1 h. After evaporation and concentration to 5 mL at 60 °C, each sample was filtered through a 0.45 μm filter, and the filtrate was used for HPLC analysis. The HPLC analysis was carried out on a Zorbax SB-C_18_ column (250 mm × 4.6 mm, 5 μm) with an extend-C_18_ guard column (12.5 mm × 4.6 mm, 5 μm). The mobile phase was acetonitrile (solvent A) and water (solvent B), with a flow rate of 1.0 mL/min and the following gradient: A:B ratios of 19:81 for 0 to 35 min; 29:71 for 35 to 70 min; 40:60 for 70 to 100 min; and 85:15 for 100 to 120 min. The sample was detected at 203 nm wavelength by UV–spectrometry.

### 4.7. RNA Isolation and Quantitative RT-PCR Analysis

RNA isolation and quantitative RT-PCR analysis were conducted as described by Lei et al. [13]. Four-week-old ginseng seedlings, grown in normal conditions, were treated with 30 μM ABA for 3 h. Total RNA was extracted with a RNeasy Plant Mini Kit (Qiagen, cat. No. 74904). A total of 4 μg DNase I-treated total RNA was used as the template for reverse transcription via M-MLV reverse transcriptase (Promega, cat. no. M170A). qRT-PCR was performed with 0.2 μL cDNAs, SYBR premix ExTaq, and gene-specific primers via a 7300 Real-Time PCR system. The reaction conditions included 40 cycles at 94 °C for 5 min, 94 °C for 15 s, and 60 °C for 34 s.

## Figures and Tables

**Figure 1 ijms-24-09194-f001:**
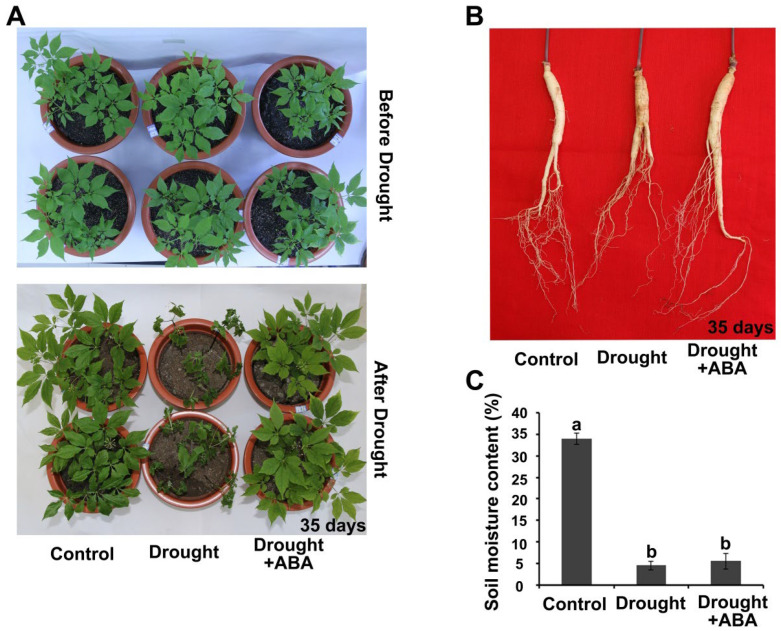
Drought resistance is enhanced by ABA treatment in *Panax ginseng* C. A. Meyer. (**A**) Morphological changes in *Panax ginseng* leaves after drought or drought + ABA treatment for 35 days. Dormancy-broken *Panax ginseng* were grown under normal conditions for 4 weeks, and then the water supply was stopped for 35 days, and at same time, 15 μM ABA was sprayed on the leaves every two days as drought + ABA treatment. (**B**) Morphological changes in *Panax ginseng* roots after drought or drought + ABA treatment for 35 days. *Panax ginseng* materials are treated as in A. (**C**) Soil relative water content was measured after stopping the water supply for 35 days. Letters indicate significant difference at the level of *p* < 0.05. Three independent trials were conducted with similar results in this figure.

**Figure 2 ijms-24-09194-f002:**
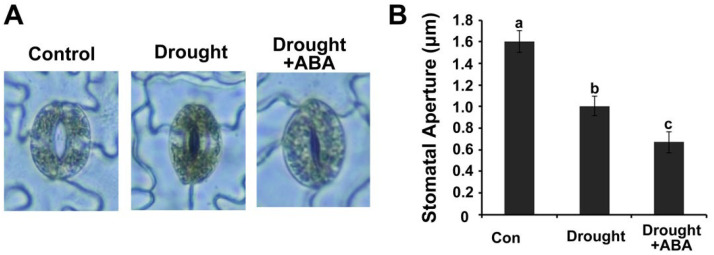
Stomatal closure of *Panax ginseng* leaves under drought stress is promoted by ABA treatment. (**A**) Morphology of stomata in *Panax ginseng* leaves after drought or drought + ABA for 28 days. Dormancy-broken *Panax ginseng* were grown under normal conditions for 4 weeks, and then the water supply was sopped for 28 days, and at same time, 15 μM ABA was sprayed onto the leaves every two days as drought + ABA treatment. The epidermal strips of *Panax ginseng* leaves were taken off and the stomata were observed. Sixty stomata were measured form one sample in each replicate. Three independent experiments were conducted with similar results. (**B**) Measurement of stomatal aperture in *Panax ginseng* leaves. Letters indicate significant difference at the level of *p* < 0.05.

**Figure 3 ijms-24-09194-f003:**
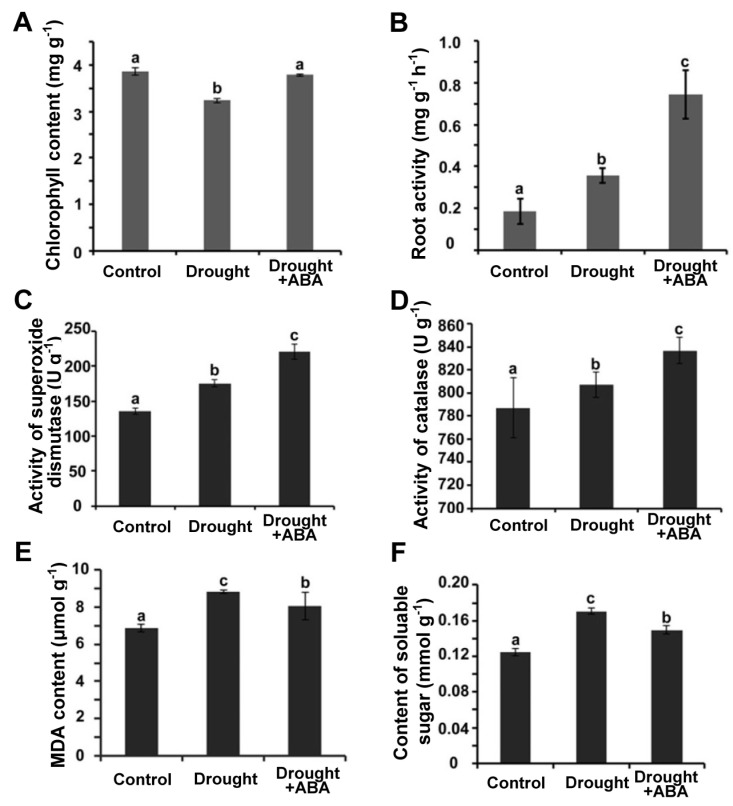
Physiological response of *Panax ginseng* to ABA treatment under drought stress. (**A**) Chlorophyll content of *Panax ginseng* leaves grown under drought and drought + ABA conditions. (**B**) Roots activity of *Panax ginseng* grown under drought and drought + ABA conditions. (**C**) Superoxide dismutase activity of *Panax ginseng* leaves grown under drought and drought + ABA conditions. (**D**) Catalase activity of *Panax ginseng* leaves grown under drought and drought + ABA conditions. (**E**) Malondialdehyde (MDA) content of *Panax ginseng* leaves grown under drought and drought + ABA conditions. (**F**) Soluble sugar content of ginseng leaves grown under drought and drought + ABA conditions. In this figure, dormancy-broken *Panax ginseng* was grown under normal conditions for 4 weeks, and then the water supply was stopped for 35 days, and at same time, 15 μM ABA was sprayed onto the leaves every two days as drought + ABA treatment. After 35 days, the chlorophyll content, roots activity, SOD activity, CAT activity, MDA content, and soluble sugar content were measured. Values are means ± S.D. of three biological repetitions. Letters indicate significant difference at the level of *p* < 0.05. Three independent experiments were performed with similar results.

**Figure 4 ijms-24-09194-f004:**
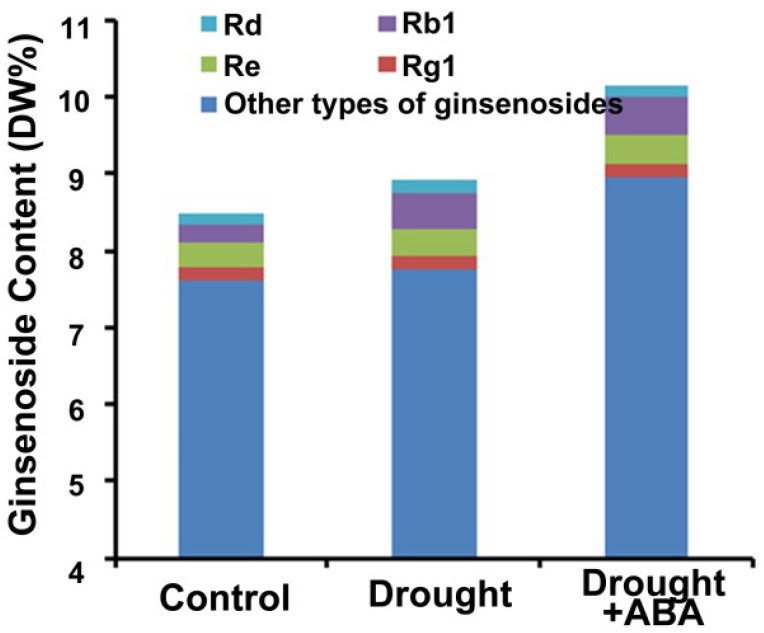
The accumulation of ginsenoside is promoted by ABA treatment in *Panax ginseng* roots under drought conditions. After drought and drought + ABA treatment for 35 days, ginsenoside content was determined by HPLC.

**Figure 5 ijms-24-09194-f005:**
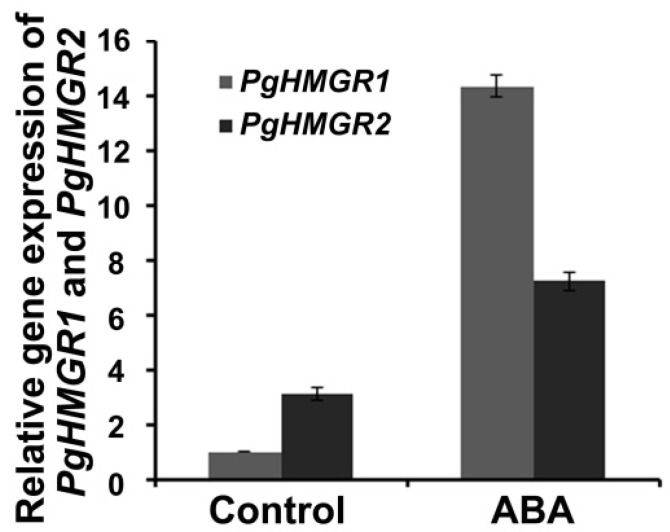
Expression of *PgHMGR1 and PgHMGR2* is induced by ABA treatment in *Panax ginseng* leaves. RNAs were isolated from *Panax ginseng* leaves treated with 30 μM ABA for 3 h. Three independent experiments were performed, each with three replicates.

## Data Availability

Data presented here are available on request from correspondence.

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
