# Peer review of "Drought Resistance and Ginsenosides Biosynthesis in Response to Abscisic Acid in Panax ginseng C. A. Meyer"

_ijms, 2023, doi:10.3390/ijms24119194_

Round 1
Reviewer 1 Report
In my opinion, the work has many flaws and errors. The introduction without need describes the entire signal transduction pathway involved in the action of ABA. This is not the subject of research. It is not related to the purpose of the work, which is not clearly presented and justified. It was necessary to state why ABA was used and what the working hypothesis is and justify it. Why only one concentration of ABA was used?
The terminology drought tolerance is incorrectly used instead of drought resistance, which is the effect of avoiding/delaying the state of stress (dehydration), tolerance of the state of stress.
The level of the leaf water content (RWC) indicating the level of dehydration (state of stress), was not determined. Moreover, these results do not bring anything new.
It was shown that ABA is involved in activating the stress avoidance mechanism, i.e. stomatal closure and stress tolerance, i.e. activation of the antioxidant system. The level of leaf hydration (RWC) was not determined which is a mistake.
ABA has been shown to activate the synthesis of ginsenosides but how does this relate to drought resistance?
​
Author Response
Review 1 # Comments and Suggestions for Authors
In my opinion, the work has many flaws and errors. The introduction without need describes the entire signal transduction pathway involved in the action of ABA. This is not the subject of research. It is not related to the purpose of the work, which is not clearly presented and justified. It was necessary to state why ABA was used and what the working hypothesis is and justify it.
- Response: We fully agree with the Reviewer. We have rewritten the introduction section and removed the ABA signaling pathway and included the working hypothesis in the revised manuscript.
Why only one concentration of ABA was used?
- Response: Many thanks. We have searched relevant references and did experiments with different concentration of ABA, and finally employed the treatment (15uM ABA). This ABA concentration is widely accepted for leaf spraying (see reference 27 and 28).
The terminology drought tolerance is incorrectly used instead of drought resistance, which is the effect of avoiding/delaying the state of stress (dehydration), tolerance of the state of stress.
- Response: We thank the Reviewer for this excellent suggestion. The terminology drought resistance instead of drought tolerance was employed in the revised manuscript.
The level of the leaf water content (RWC) indicating the level of dehydration (state of stress), was not determined. Moreover, these results do not bring anything new. It was shown that ABA is involved in activating the stress avoidance mechanism, i.e. stomatal closure and stress tolerance, i.e. activation of the antioxidant system. The level of leaf hydration (RWC) was not determined which is a mistake.
- Response: Many thanks. Leaf water content (RWC) is employed to quantify the degree of leaf water insufficiency or unsaturation in many model and crop plants. However, different plant tissues can have different values of absolute water content (in grams per mass unit) even at full saturation (see reference 29). In this study, we focus on the root, a highly valued tissue in Panax ginseng, instead of leaf. We examined the soil moisture content (Figure 1C) and analyzed the root growth (Figure 1B) to show the drought stress and its effect on the root growth, and these methods have been widely employed in other drought research in Panax ginseng (see reference ). In addition, we measured some leaf characteristics like chlorophyll content and stomatal closure to show the leaf response to drought stress in Panax ginseng.
ABA has been shown to activate the synthesis of ginsenosides but how does this relate to drought resistance?
- Response: Thank you very much for this excellent suggestion. The potential relationship among ABA, ginsenosides biosynthesis and drought resistance have been discussed in the revised discussion section
Reviewer 2 Report
This is an interesting study in the field of plant physiology and secondary metabolite accumulation.
However, there have three main points which the authors have to address before further consideration.
1. The manuscript need to be improved and avoid similarity (I check more than 25% similarity
2. Why only use 1 concentration of ABA? Do you have done any experiment before? Which is reference?
It is too hard to conclude supplemented ABA is help the plant overcome the drought stress and improve of ginsenoside with only one treatment (15uM ABA).
3. HMGR is one of the first enzyme in ginsenoside synthesis pathway. P. ginseng squalene synthase (PgSS2), P. ginseng squalene epoxidase (PgSE2), P. ginseng protopanaxadial synthase (PgPPDS) and P. ginseng protopanaxatriol synthase (PgPPTS) is more important. Why don’t you do gene expression for them. Please refer at Murthy et al 2014 (Ginsenosides: prospective for sustainable biotechnological production. Appl Microbiol Biotechnol (98:6243–6254) and Lee et al 2018 (Cell culture system versus adventitious root culture system in Asian and American ginseng: a collation. Plant Cell Tiss Organ Cult (2018) 132:295–302)
Minor comments:
- "Panax ginseng" should be in italic, please check and correct
- Line 111-113: "The epidermal strips of ginseng leaves.........similar results" should be moved to material and method parts.
- Line 141-145: "In this figure, .........soluble content were measure" should be moved to material and method parts.
- Line 148: What is MOD mean? Do you mean MDA? Please check and correct.
- Section 4.4. I didn't see method for MDA content and soluble sugar content. Please detail the protocol for them.
- Please cite the reference for the protocol at section 4.6; 4.7.
.
Should be improved.
Author Response
Review 2 # Comments and Suggestions for Authors
This is an interesting study in the field of plant physiology and secondary metabolite accumulation.
- Response: Thank you very much for these encouraging comments. We have made extensive revision of this manuscript according to referees’ comments. Hopefully, this version could meet the standard for publication.
However, there have three main points which the authors have to address before further consideration.
- The manuscript need to be improved and avoid similarity (I check more than 25% similarity)
- Response: Many thanks. We have extensively revised this manuscript to avoid similarity.
- Why only use 1 concentration of ABA? Do you have done any experiment before? Which is reference? It is too hard to conclude supplemented ABA is help the plant overcome the drought stress and improve of ginsenoside with only one treatment (15uM ABA).
- Response: Thank you very much. We have searched relevant references and did experiments with different concentration of ABA, and finally employed the treatment (15uM ABA). This ABA concentration is widely accepted for leaf spraying (see reference 27 and 28 ).
- HMGR is one of the first enzyme in ginsenoside synthesis pathway. P. ginseng squalene synthase (PgSS2), P. ginseng squalene epoxidase (PgSE2), P. ginseng protopanaxadial synthase (PgPPDS) and P. ginseng protopanaxatriol synthase (PgPPTS) is more important. Why don’t you do gene expression for them. Please refer at Murthy et al 2014 (Ginsenosides: prospective for sustainable biotechnological production. Appl Microbiol Biotechnol (98:6243–6254) and Lee et al 2018 (Cell culture system versus adventitious root culture system in Asian and American ginseng: a collation. Plant Cell Tiss Organ Cult (2018) 132:295–302)
- Response: Many thanks. These relevant references have been cited in the revised manuscript. ABA was shown to regulate HMGR gene promoter and ginsenoside production in Panax quinquefolium hairy root cultures, and HMGR was examined in this study for consistence (see reference ).
Minor comments:
- "Panax ginseng" should be in italic, please check and correct
- Response: Many thanks. the mistake has been corrected in the revised version.
- Line 111-113: "The epidermal strips of ginseng leaves.........similar results" should be moved to material and method parts.
- Response: Yes, this sentence has been moved to material and method parts.
- Line 141-145: "In this figure, .........soluble content were measure" should be moved to material and method parts.
- Response: Many thanks. This sentence has been moved to material and method parts.
- Line 148: What is MOD mean? Do you mean MDA? Please check and correct.
- Response: We thank the Reviewer for identifying this typo and it has been corrected in the revised revision.
- Section 4.4. I didn't see method for MDA content and soluble sugar content. Please detail the protocol for them.
- Response: Many thanks. Methods for measuring MDA content and soluble sugar content have been included in the revised material and method section.
- Please cite the reference for the protocol at section 4.6; 4.7.
- Response: Yes, references for the protocol at section 4.6; 4.7 have been cited in the revised version.
Round 2
Reviewer 1 Report
The work has been improved. However, the manuscript has still some flow.
In my opinion, it still needs language correction. In some parts, English is difficult to understand.
Moreover, I noticed some terminology errors, e.g. line 237 resistance, not tolerance, line 37 - change drought adaptation to drought adjustment
Lines 15-16 - strange sentence, incomprehensible.
Line 251 ….irregular arranged irregularlyment- …. What does it mean?
Lins 88-89 .... we examined the response of drought resistance and ginsenosides biosynthesis to ABA in Panax ginseng - it sounds strange
Authors should carefully check the manuscript and rewrite removing redundant parts.
Reviewer 2 Report
The authors have revised manuscript carefully.